# Acute Rehabilitation following Traumatic anterior shoulder dISlocAtioN (ARTISAN): protocol for a multicentre randomised controlled trial

Rebecca Samantha Kearney [ORCID],[1] Gurmit Dhanjal,[1] Nicholas Parsons,[1] David Ellard [ORCID],[1] Helen Parsons,[1] Aminul Haque,[1] Eleni Karasouli,[1] James Mason,[1] Henry Nwankwo,[1] Jaclyn Brown,[1] ZiHeng Liew,[1] Stephen Drew,[2] Chetan Modi,[2] Howard Bush,[2] David Torgerson,[3] Martin Underwood[1]

► Prepublication history and additional materials for this paper is available online. To view these files, please visit the journal online (http://dx.doi.org/10.1136/bmjopen-2020-040623).

¹Warwick Medical School, University of Warwick, Coventry, UK
²Trauma and Orthopaedics, University Hospitals Coventry and Warwickshire NHS Trust, Coventry, UK
³York Clinical Trials Unit, The University of York, York, UK

**Correspondence to**
Dr Rebecca Samantha Kearney;
R.S.Kearney@Warwick.ac.uk

## ABSTRACT

**Introduction** First-time traumatic anterior shoulder dislocation (TASD) is predominantly managed non-operatively. People sustaining TASD have ongoing pain, disability and future risk of redislocation. There are no published randomised controlled trials (RCTs) comparing different non-operative rehabilitation strategies to ascertain the optimum clinically effective approach after TASD.

**Methods and analysis** In this multicentre adaptive RCT, with internal pilot, adults with a radiologically confirmed first time TASD treated non-surgically will be screened at a minimum of 30 sites. People with neurovascular complications, bilateral dislocations or are unable to attend physiotherapy will be excluded.

Randomisation will be on a 1:1 treatment allocation, stratified by age, hand dominance and site. Participants will receive a single session of advice; or a single session of advice plus offer of further physiotherapy (maximum 4 months). The primary analysis will be the difference in Oxford Shoulder Instability Score at 6 months. A sample size of a minimum of 478 participants will allow us to show a four point difference with 90% power.

An embedded qualitative study will explore the participants' experiences of the trial interventions.

**Ethics, registration and dissemination** Funded by NIHR HTA (16/167/56), 1 June 2018; National Research Ethic Committee approved (18/WA/0236), 26 July 2018. First site opened 5 November 2018 and final results will be updated on trial registries and submitted to a peer-reviewed journal and will inform rehabilitation strategies after a TASD. Study Within A Trial (SWAT) funded by MRC (MR/R013748/1), 1 May 2019; registered on the MRC-HTMR All-Ireland Hub (reference number SWAT 121).

**Trial registration number** ISRCTN63184243. (Trial stage: Pre-results)

## INTRODUCTION

The shoulder is the most frequently dislocated joint; occurring in 8.2–23.9 per 100 000 people per year.[1] Traumatic anterior shoulder dislocations (TASDs) occur when excessive forces during a traumatic event displace the humeral head frontwards, out of the shoulder socket, resulting in the joint surfaces completely losing contact.[1–3] They predominantly occur in men under 25 years during high-impact events and women over 80 years during low-impact events.[1]

People sustaining TASD often have ongoing pain, disability and morbidity linked to redislocation and subsequent need for repeated management.[1–3] Rehabilitation may reduce these events and restore a functional, painless shoulder through restoration of movement and retraining muscles.[2] However, a Cochrane review has found no evidence to support this[2]; Dutch national guidelines explicitly do not refer to physiotherapy[3] and UK guidelines cite referral 'may be helpful'.[1]

This research protocol will investigate the hypothesis regarding the nature and extent of what physiotherapy is required for the management of patients following TASD with an embedded 'Study Within A Trial (SWAT)' to compare two strategies for retention of participants in randomised controlled trial (RCTs).

## RESEARCH QUESTIONS

### RCT research question

In adults with first time TASD managed non-operatively does a single session of advice and course of physiotherapy improve the Oxford Shoulder Instability Score (OSIS) by

a minimum of four additional points at 6 months when compared with a single session of advice only?

### SWAT research question

Does making a courtesy introductory telephone call to recruited participants improve response rates to 6 months follow-up questionnaires compared with a written card with similar information?

## OBJECTIVES

### RCT primary objective

To quantify and draw inferences on observed differences in the OSIS score between a single session of advice and physiotherapy with a single session of advice only, for adults with first time TASD managed non-operatively, 6 months postrandomisation.

### RCT secondary objectives

► To estimate comparative cost-effectiveness (cost/quality-adjusted life-years, QALY) of the two trial treatments, from an National Health Service (NHS) and personal social services perspective.
► To determine the difference in complication rate (eg, shoulder redislocation) in the first 12 months between the trial treatment groups.
► To quantify and draw inferences between the functional status (OSIS) of the trial treatment groups at 6 weeks, 3 and 12 months.
► To quantify and draw inferences on observed differences in the functional status (Disabilities of the Arm, Shoulder and Hand) of between the trial treatment groups at 6 weeks, 3, 6 and 12 months.
► To quantify and draw inferences on observed differences of health-related quality of life (EQ-5D-5L) between the trial treatment groups at 6 weeks, 3, 6 and 12 months.
► To qualitatively explore participants' experiences of receiving the trial treatments and identify facilitators and obstacles to adhering to them.

### SWAT primary objective

Evaluate the impact of making a courtesy call to newly recruited participants in the Acute Rehabilitation following Traumatic anterior shoulder dISlocAtioN (ARTISAN) trial on response rates to 6 months follow-up questionnaires compared with a written card with similar information.

### SWAT secondary objectives

► Time to response to the questionnaires at each time points.
► Response rates at 6 weeks, 3, 6 and 12 months.
► Completeness of responses.
► Number of reminder notices required.
► Cost of intervention per participant.

## METHODS AND ANALYSIS

### Study design

Multicentre, adaptive, RCT with internal pilot and SWAT.

### Outcome measures

Outcome measures will be collected at baseline (prerandomisation) at the recruiting site and then at 6 weeks, 3, 6 and 12 months centrally from Warwick Clinical Trials Unit (WCTU) by postal questionnaire, mobile application, telephone or email.

### RCT primary outcome

OSIS: Self-completed outcome measure containing 12 questions (0–4 points each), with possible scores from 0 (best function) to 48 (worst function).[4 5] Questions relate to activities of daily living particularly relevant to patients exhibiting shoulder instability and has been designed to assess outcome of therapy.

### RCT secondary outcomes

#### Disabilities of the Arm, Shoulder and Hand

Self-completed shortened version of the Disabilities of the Arm, Shoulder and Hand (QuickDASH) questionnaire. Uses 11 items to measure physical function and symptoms in people with any or multiple musculoskeletal disorders of the upper limb. Designed to help describe the disability experienced by people with upper-limb disorders and monitor changes in symptoms and function over time.[6]

#### EQ-5D-5L

Health-related quality of life measure consisting of five dimensions, each with 5 levels of response. Each combination of answers is converted into a health utility score. Gives a single preference based index value for health status that can be used for broader cost-effectiveness comparative purposes.[7]

Complications: Complications will be defined into three categories:
1. Predefined complications directly related to the interventions.
2. Predefined complications directly caused by the primary TASD event not identified by the initial assessing clinician, but subsequently identified.
3. Complications not related to the intervention or TASD event and will subsequently not be formally analysed or reported.

#### Resource use questionnaires

Primary health economic analysis will concentrate on direct intervention and healthcare/personal social services costs, while wider impact (societal) costs will be included within the sensitivity analyses.

#### Embedded qualitative interviews

A purposive sample, up to 50 participants, informed by treatment allocation, gender, age and outcome will be invited for one interview, to explore the participant experience of receiving the interventions and facilitators and obstacles to adhering to them.

## SWAT primary outcome

Response rate: Proportion of participants who return the questionnaire.

## SWAT secondary outcomes

Time to response: Date of first sending questionnaire to date of questionnaire received by study team.

Completeness of responses: Number of missing items per questionnaire.

Number of reminder notices: Number of notices administered by the trial team.

Cost of intervention: Cost of phone call or postcard per participant.

## Sample size

### RCT sample size

A worthwhile difference from this intervention is four points on the OSIS.[8 9] The SD of the OSIS 6 months after injury is 10 points.[8 9] However, the literature has predominantly included a young population, given that we will recruit a wider range of ages a larger SD is expected, therefore an SD of 12 is more likely. To show a four-point difference at 5% significance level with 90% power allowing a margin of 20% loss during the follow-up needs a minimum of 478 participants.[5 9]

### SWAT sample size

The sample size for the SWAT is limited by the host trial sample size. The SWAT opened to recruitment after the internal pilot (50 participants), therefore, approximately 428 participants (214 per group) are expected and would provide 90% power and 5% significance to detect differences in retention rates of approximately 11%.

## Screening and eligibility

### RCT screening and eligibility

Adults presenting at trial sites (NHS hospitals) with a first-time TASD, confirmed radiologically, managed non-operatively (decided by the treating clinician and patient) will be identified as potential participants. The site team will undertake eligibility checks and details will be entered on a monthly screening log.

Participants will be excluded if they have significant neurovascular complications; bilateral dislocations; have been randomised previously; unable to adhere to trial procedures or unable attend physiotherapy within 6 weeks of injury.

All eligible participants will be provided with verbal and written information about the study. If eligible and willing to join the study, a consent form will be signed (see online supplemental material), followed by collection of baseline demographic data, preinjury and postinjury functional outcomes.

### SWAT screening and eligibility

Participants recruited into ARTISAN, who consent to being contacted by multiple methods, will be eligible for the SWAT.

## Randomisation

### RCT randomisation

Participants will be randomised at the end of the initial advice session via a web-based system. Allocation concealment will be maintained by an independent randomisation team at WCTU, who will be responsible for generation of the sequence and will have no role in the allocation of participants. The treatment group will be allocated by computer using a minimisation algorithm with a random element and stratification by participant age (39 years old and under and 40 years old and over), hand dominance and recruiting site.

### SWAT randomisation

Minimisation with a random factor will be used to avoid imbalance between the SWAT intervention arms. The allocation ratio will be 1:1 and stratified by the ARTISAN allocation arm. Allocation concealment will be maintained by an independent randomisation team who will be responsible for generation of the sequence and have no role in the allocation of participants.

### Postrandomisation withdrawals

Participants may discontinue at any time without prejudice. Unless a participant explicitly withdraws consent, data will be collected until the end of the trial. For participants explicitly withdrawing consent for follow-up procedures, trial data obtained up until the point of withdrawal will be included. Participants will have the option to withdraw from the trial-related questionnaires, but continue to provide routine NHS data for trial purposes. Participants who withdraw will not be replaced. Participants may be withdrawn from the trial at the discretion of the investigator and/or trial steering committee for safety concerns.

## Interventions

### RCT interventions

The interventions will be reported in line with the template for intervention description and replication (TIDIER) and Consensus on Exercise Reporting Template (CERT) checklists.[10 11] All participating centres will receive an initial training session from the trial team. Following this, a lead physiotherapist at each site will be identified to complete subsequent training of additional physiotherapists. This training will be supported with web-based materials and a trial intervention manual.

All participants will receive a period of initial immobilisation in internal rotation as per UK national guidelines, for a duration of up to 2 weeks from date of injury.[1] At this time participants will be provided with a web-link to phase 1 of the ARTISAN advice materials and provided with a paper-based booklet version covering:

► What has happened to me?
► What can go wrong?
► How do I stop this happening again?
► How long do I have to wear my sling?
► Should I move my arm?
► How do I control my pain?

► When can I return to usual activities?

► What if something goes wrong?

All participants will subsequently receive an appointment for a physiotherapy advice session within 6 weeks of injury. At the first appointment, all participants will receive a single session of advice to aid self-management, lasting up to 1 hour and administered by an ARTISAN trained physiotherapist. Following routine assessment, the physiotherapist will deliver the above core set of intervention components and in addition:

► Points of contact if complications occur or expected recovery times are not achieved.

► A core set of progressive phase 2 range of movement exercises and what they aim to achieve.

► Enhancing self-management behaviours through the addition of goal setting, exercise planning and diaries.

The physiotherapist will provide details of web-based materials, which will include all the core components above in written and video format, and will include a dedicated area for participants to set goals and keep diaries. The physiotherapist will also inform the participants that the website resources contain phase 3 progressive strengthening exercises and what they aim to achieve and later stage phase 4 progression to aid return to sports. Participants will be offered paper-based alternatives.

Following completion of the first physiotherapy appointment, the participant will be randomised, allocating them to this advice session alone or to this advice session plus the offer of additional physiotherapy.

Participants randomised to the advice session only, will only receive this physiotherapy intervention. However, if they are experiencing ongoing issues which are not improving, as part of the control intervention, the advice is for the participant to contact the clinic they attended or see their general practitione. If a participant follows this advice and self-initiates further physiotherapy, this will not be deemed a deviation from the protocol. However, if someone other than the participant initiates further physiotherapy (another clinician) then this will be deemed a protocol deviation. This data will be collected as part of the trial dataset.

Participants randomised to receive additional physiotherapy will be offered of at least one additional physiotherapy session. Each additional session will last for up to 30 min, over a maximum duration of 4 months from date of randomisation. The number and frequency of any additional sessions will be jointly agreed between the clinician and participant, in keeping with standard practice. The course of physiotherapy will involve teaching and supervising the 'core set' of progressive exercises offered to the control arm in addition to being able to tailor treatment as per their usual practice.

### SWAT interventions

Participants will be randomised to receive either:

1. An introductory call within 2 weeks of being randomised to include thanking participants; reminding them of their valuable contribution and that they will be contacted at 6 weeks and 3, 6 and 12 months; informing them of when the trial results are expected and to contact the ARTISAN team if they have queries.

2. A postcard-sized handwritten card, with similar content as above, signed by the chief investigator and trial manager, posted to participants within 1 week of being randomised.

### Fidelity and quality assurance of interventions

The trial team will implement a standardised approach of evaluating fidelity[12] of direct observations and audio recordings (twice annually); and self-reporting of the trial interventions. Any issues will be discussed by the trial management group (TMG), who will recommend appropriate action such as feedback to sites and additional training. If issues are not resolved following recommendations they will be escalated to oversight committees.

During site quality assurance visits the research teams will also be asked to provide radiological confirmation of the participants initial shoulder dislocation. If evidence is not found, participants will be withdrawn from the main trial sample, but will continue follow-up procures and retained as part of a pre-planned sensitivity analysis.

### Blinding

As the type of rehabilitation used will be clear to the participant, they cannot be blind to treatment. The treating clinician will also not be blind, but will take no part in follow-up processes, these will be managed centrally from the WCTU to reduce the assessment bias.

### Adverse event management

Serious adverse events (SAEs) will be entered onto the SAE reporting form and reported to the central study team. SAEs that may be expected will be predefined and recorded on the participant's follow-up questionnaire. SAEs that may be expected as part of a TASD are: damage to nerve or blood vessels, fractures, redislocation, torn ligaments or muscles, persistent exacerbation of shoulder pain, restriction of range of movement, adhesive capsulitis (frozen shoulder) and persistent instability. All SAEs will be reported to the sponsor (Warwick University and University Hospitals Coventry and Warwickshire), ethics committee and oversight committees.

### End of trial

The trial will end when all participants have completed their 12-month follow-up. The trial will be stopped prematurely if mandated by the Ethics Committee; following recommendations from relevant oversight committees; or funding ceases.

### Trial oversight

The TMG will meet monthly and report to the independent trail steering committee (TSC) and data monitoring and safety committee (DMSC). The TSC and DMSC will hold meetings annually and be responsible for approval of the protocol, monitoring and supervising progress.

## Patient and public involvement

The clinical team consulted with patients when developing this project and subsequently we had a meeting with a group representing our target participants. We discussed experiences and expectations of services and plans for the trial. Service users' perspectives were key in the development of this protocol. Patient and public involvement (PPI) membership is represented on TMG and TSC committees and have reviewed all study materials. PPI members of the team will contribute to dissemination on trial completion.

## Analysis

### RCT statistical analysis

The internal pilot will inform and test recruitment strategies for the main trial in 12 sites over 6 months. A recruitment rate of one participant per month per centre was estimated for the 6 months pilot phase, with 50 participants in total. The decision to progress to the main trial will be made based on predefined progression criteria. The criteria for continuing will be achievement of 75%–100% of the recruitment target; criteria for the need to review and amend trial procedures before continuing will be achieving 50%–75% of the target; criteria for not proceeding will be achieving less than 50% of the target. These randomised participants will be retained in the final analysis.

A detailed statistical analysis plan which will be agreed with the DMSC prior to the analyses taking place. All data will be reported in accordance with the Consolidated Standards of Reporting Trials (CONSORT) statement. Baseline data will be summarised to check comparability between treatment arms, and screening data will be checked to highlight any characteristic differences between those individuals in the study, those ineligible and those eligible but withholding consent.

The research team recognise the theoretical possibility of therapist effects. Here, we are randomising by individual; it is the same therapist who will be seeing people in both arms of the trial, and the treatment is individually delivered ensuring there are no group effects. To keep presentation of key information as standard throughout the study, the initial therapist contact will be described in the study manual and will use supporting media. We intend to recruit from a minimum of 30 sites where there will be multiple therapists taking part over the course of the study (eg, sickness and holiday coverage). This gives the number of estimated therapist level clusters upwards of 60 and potential cluster sizes of around four participants in each arm of the trial. With such small cluster sizes, even if there were to be important therapist effects, their effect on statistical power would be minimal.

To address the possibility that these might exist, a single interim analysis is preplanned to re-estimate the sample size. This will occur after approximately 200 participants have completed the 3 months follow-up questionnaire, while recruitment is still open. The pooled SD of the primary outcome and presence of therapist effects will be estimated by calculating the therapist intraclass correlation coefficient and its 95% CI. Models will not include treatment effects. A revised sample size calculation will be discussed with DMC and TSC and if appropriate, permission requested from the funder to adjust the sample size.

All primary analyses are planned to be on an intention-to-treat basis with secondary per protocol analyses. The main analysis will investigate differences in the primary outcome measure, 6 months after randomisation, between the two treatment groups. Unadjusted and adjusted regression analyses will be used to estimate the between group difference. The adjusted analyses will adjust for the stratification variables, baseline scores and any other clinically important variables. Since individual clinicians will treat only a small number of participants enrolled in the trial, we do not expect clinician-specific effects to be important in this study.

Descriptive statistics of functional outcome data (ie, QuickDash and EQ-5D-5L) at each time point will be constructed with between group analyses following the method set out for the primary analysis above. Patterns of recovery will also be explored. Secondary analyses will include $\chi^2$ tests to compare the number of dislocations and other complications between allocation groups. For important complications (eg, dislocations), Kaplan-Meier curves of the time to complication will be constructed. If sufficiently large numbers of complication groups are observed, Cox regression models will be constructed to compare the time to complication in each arm. Temporal effects will be investigated using a multilevel model of all follow-up data.

Two prespecified subgroup analyses will be undertaken to assess whether there is evidence of differing intervention effect: hand dominance (injured shoulder dominant arm vs injured shoulder is non-dominant arm and age (younger participants vs older participants). Because of the small sample size these analyses will be exploratory. The subgroup analyses will follow the methods described for the primary analysis, with additional interaction terms incorporated into the mixed-effects regression model to assess the level of support for these hypotheses. We will additionally present the main result separately for younger and older age groups for the benefit of future systematic reviewers who may be focused on different age groups. We will draw no inference from these analyses.

Careful monitoring of missingness and crossovers will be conducted. If judged appropriate, multiple imputation will be used to account for missing data, with all necessary assumptions reported. If large numbers of treatment crossovers are observed, Complier-Average Causal Effect models will be used.

### RCT health economic analysis

A prospectively planned analyses will be detailed in a health economic analysis plan, agreed with the DMSC. Analysis will be conducted from an NHS and personal social services perspective, with methods adhering to the

recommendations of the National Institute for Health and Care Excellence (NICE) reference case.[13]

Costs of the intervention groups will be estimated, reflecting resource inputs associated with rehabilitation and broader healthcare resource utilisation. Resource use associated with the index condition will be captured through routine health service data collection systems and participant questionnaires administered at each follow-up time point.

Unit costs will be estimated from local and national sources using established accounting methods, reflated to current prices. Health-related quality of life will be measured at baseline and at all follow time points using the EQ-5D-5L measure. Responses will be used to generate QALYs using the UK value set recommended by NICE guidance at the time of analysis.[14]

Within-trial analysis using bivariate regression of costs and QALYs, with multiple imputation of missing data, will inform a probabilistic assessment of incremental treatment cost-effectiveness. Missingness mechanisms will be explored and multiple imputation methods will be used where appropriate to avoid biases associated with complete case analysis. Findings will be analysed and visualised in the cost-effectiveness plane, as cost-effectiveness acceptability curves, net monetary benefit and value of information analysis. Costs and outcomes arising during the trial will be undiscounted, reflecting the 12-month time horizon. Sensitivity analyses will be undertaken to explore uncertainties and to consider issues of generalisability of the study.

### SWAT analysis

All eligible participants will be included on an intention-to-treat basis, using two-sided statistical significance at the 5% level. We will summarise baseline characteristics of participants by the type of SWAT intervention sent. An average cost per participant will be estimated for each SWAT intervention arm. For the primary outcomes of questionnaire response rates, the difference in proportions will be calculated with 95% CIs, and the primary analysis will be a $\chi^2$ test to assess statistical association. Additionally, a logistic regression adjusting for age, gender and host trial treatment allocation will be performed to investigate the effects of these variables. A per-protocol analysis will also be performed.

The secondary outcome of time to questionnaire return will be assessed by a Kaplan-Meier curve and the SWAT interventions compared by log rank test. Cox regression will be applied adjusting for participant age, gender and host trial treatment allocation, and the effect of the intervention reported. The requirement for any questionnaire return reminder will be analysed in the same way as the primary outcome).

### Embedded qualitative study

Qualitative data will be analysed using the Framework method,[15] which broadly includes data familiarisation; identifying a thematic framework; indexing; charting; mapping and interpretation. NVivo software will be used to facilitate this process. Researcher bias will be minimised through regular cross-checking of data and findings by members of the research team. Transcripts will be returned to participants (where necessary) providing them with the opportunity to check the transcripts for accuracy and authenticity and to offer any subsequent reflections. Quotes will be used as exemplars of key points in the writing up of these data. These qualitative data will be used to provide insight into participant's experiences of the interventions to provide context to the quantitative results.

### ETHICS AND DISSEMINATION

This study was funded by NIHR HTA (16/167/56), 1 June 2018; National Research Ethic Committee approved (18/WA/0236), 26 July 2018. The first site opened to recruitment 5 November 2018. The SWAT was funded by MRC (MR/R013748/1), 1 May 2019 and registered on the MRC-HTMR All-Ireland Hub (reference number SWAT 121).

The trial will be conducted in full conformance with the principles of the Declaration of Helsinki and to Good Clinical Practice guidelines and comply with all applicable UK legislation and Warwick University Standard Operating Procedures. The trial will be reported in line with the CONSORT statement.[16]

The results of this project will be disseminated through peer-reviewed journals, conference presentations among the orthopaedic and rehabilitation networks, policymakers, patient-specific newsletters and local mechanisms at all participating centres.

### MAIN STRENGTHS AND LIMITATIONS

This is the first UK-wide multicentre RCT across a minimum of NHS 30 sites. The primary outcome measure is patient centred. The main limitation is that the intervention could not be blinded.

**Acknowledgements** The research team would like to acknowledge Alwin McGibbon and Matt Kennedy for their contribution throughout the development and main trial as a PPI Representatives and Phil Moss for data clerk expertise.

**Contributors** RSK, MU, DRE, EK, DT, SD, HB and CM wrote the background section and developed the research question. RSK, MU, DRE, EK, DT, JB, ZHL and GD wrote the research methodology and management sections of the protocol. DRE, NP, HP, AH and JM. HN wrote the sample size and analysis sections of the protocol. All authors reviewed and approved the final manuscript.

**Funding** This trial was funded by NIHR HTA (16/167/56), 01/06/18. The SWAT was supported by the Medical Research Council (MRC; grant number MR/R013748/1).

**Disclaimer** The funder has not been involved in the design of the study. The views expressed are those of the authors and not necessarily those of NIHR or the Department of Health and Social Care.

**Competing interests** RSK is a member of the UK NIHR HTA CET board, NIHR ICA Doctoral panel and previous member of the NIHR RfPB board. RSK, NP, DRE, HP, JM, MU, SD, CM and DT have all been awarded current and previous NIHR research grants. JB, GD, ZHL, AH, HN, EK and HB have none to declare. MU has received travel expenses for speaking at conferences from the professional organisations hosting the conferences. He is a director and shareholder of Clinvivo Ltd that provides electronic data collection for health services research. He is part of an

academic partnership with Serco. He was an editor of the NIHR journal series for which he received a fee.

**Patient consent for publication** Not required.

**Provenance and peer review** Not commissioned; externally peer reviewed.

**ORCID iDs**
Rebecca Samantha Kearney http://orcid.org/0000-0002-8010-164X
David Ellard http://orcid.org/0000-0002-2992-048X

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
