## [Reviewer comments · BMJ Open]

ARTICLE DETAILS

TITLE (PROVISIONAL)	Acute Rehabilitation following Traumatic anterior shoulder dISlocAtioN (ARTISAN): Protocol for a Multi Centre Randomised Controlled Trial
AUTHORS	Kearney, Rebecca; Dhanjal, Gurmit; Parsons, Nicholas; Ellard, David; Parsons, Helen; Haque, Aminul; Karasouli, Eleni; Mason, James; Nwankwo, Henry; Brown, Jaclyn; Liew, Zi Heng; Drew, Stephen; Modi, Chetan; Bush, Howard; Torgerson, David; Underwood, Martin

VERSION 1 – REVIEW

REVIEWER	Ebonie Rio La Trobe University Australia
REVIEW RETURNED	06-Jul-2020

GENERAL COMMENTS	Many thanks for the opportunity to review your manuscript outlining your trial. I have some minor suggestions for your consideration mainly around clarifying the aim as it relates to the background. I am also a little unclear in places around exposure of the physio group to physio sessions Introduction in abstract Please consider rewording "People sustaining T ASD have ongoing pain, disability, and substantial morbidity linked to high recurrence rates and subsequent need for repeated episodes of management " to link with the next sentence that seems to indicate more that we don't have these data? Or are the consequences well known and researched, we lack only those data relating to cost-effectiveness? Apologies if I have misunderstood the background. Reading the abstract - it seems the primary aim is to compare the impact of six additional physio sessions with one session? Perhaps the background could focus on whether additional access to physio improves outcomes (cost-effective is secondary aim). Please also indicate if I have not interpreted the aims correctly. Introduction Please change males to men and females to women. Please reword to investigate your hypothesis "This protocol outlines the research to provide evidence regarding the nature and extent of what physiotherapy is required for the management of patients following T ASD with an embedded 'Study Within A Trial (SWAT)' to compare two strategies for retention of participants in RCTs. Similarly - consider to compare or to investigate in terms of your primary objective.
--

	Outcome measures are well described so are timepoints. Is there blinding - will groups know about the other group? How will you define significant neurovascular complications? The randomisation after the first session is a clever design. This is a little confusing to me -is it possible you may confound your control group? "Participants randomised to the advice session only, will only receive this physiotherapy intervention. However, if they are experiencing ongoing issues which are not improving, as part of the control intervention, the advice is for the participant to contact the clinic they attended or see their GP. If a participant follows this advice and self-initiates further physiotherapy this will not be deemed a deviation from the protocol. " Do people in the physio group choose how many sessions they receive? In addition to whether participants see other clinicians, will you ask about medication use? The detail around data management and analysis is excellent. Challenges have been well considered. The early consultation with stakeholders is fantastic.
--	---

REVIEWER	Margie Olds Flawless Motion New Zealand
REVIEW RETURNED	19-Jul-2020

GENERAL COMMENTS	The authors present the problem that TASD is managed non-operatively with no published RCTs to compare non-operative strategies for this population. While this may be true, unfortunately the outcomes of this study will not improve the physiotherapy management of TASD. In the era of individualised and personalised healthcare, it is difficult to see how a single session of physiotherapy education and guidance will address the individual strength, pain, ROM and psychosocial factors of a person who has suffered a TASD. The actual information was not available for me to review, nor was it possible for me to access this information online. There are additional concerns that a single session will be compared with only 2 sessions of therapy. There are further concerns in the limitation of 6 sessions. This study therefore has no global impact as in many other countries people with a TASD receive between 16-50 sessions of therapy. I appreciate that one of the outcomes is to assess the cost effectiveness of this study. However, when compared with surgical intervention, the cost of therapy would be cheaper. My concern is that the finding of this study will present the surgical community with rationale as to why surgery should be offered to people with a FTASD, when adequate therapy has not been fully explored. Did the authors consider excluding greater tuberosity fractures? These are known to decrease risk of recurrence and may bias the findings. Similarly, will 'neurovascular complications' exclude those with axillary nerve palsy? These too will decrease rates of recurrence. Line 69- It is unclear, why the authors chose to stratify on hand dominance, compared to other risk factors that are more widely
---

	published? Age is a known risk factor and all the evidence is in agreement. There is little agreement of hand dominance. Line 251 – it is unclear who decides whether the patient will be managed non-operatively? Is there the possibility of bias because this is not clearly defined? Line 304 – it is unclear how long participants will receive immobilisation? I assume this is in internal rotation but this is not made clear. Line 342- it will be interesting to see if patients initiate further physiotherapy, if they perceive that advice and generalised exercises are all that physiotherapy has to offer? Line 346 – Again concern that 1 session will be compared to 2 sessions of therapy – Was it not possible to only have the option of 1 vs. 6? Line 349 – I am unable to access the ‘core set ‘ of generalised shoulder exercises offered to the control arm, and so cannot comment in any great depth here. In conclusion, the authors are to be commended for undertaking a study to further examine the role of physiotherapy in people with T ASD. Unfortunately it is unlikely that this study will have any global impact, as it does not serve to improve the standards of physiotherapy offered to this population. Overall, it looks to diminish the role of physiotherapy to education only. While education is an important component of physiotherapy in this population, this study is unlikely to improve the outcomes of people who receive physiotherapy for a T ASD.
--	--

REVIEWER	Daniel Cury Ribeiro University of Otago, New Zealand
REVIEW RETURNED	18-Aug-2020

GENERAL COMMENTS	Thank you for inviting me to review this interesting protocol. Below, I present some feedback and suggestions for authors to improve clarity of the reporting. Best regards, Dan Ribeiro Page 4, lines 133 to 135: should you simplify and state you are assessing the effectiveness of physiotherapy intervention plus advice compared to advice alone? Page 4, line 134, “physiotherapy with a single session of advice only”: I wonder whether this should be re-worded. The term “physiotherapy” refers to a profession. You are referring here to physiotherapy intervention/exercise therapy + advice. The way it is written may be misleading, as it seems it is a single physiotherapy session with advice only, when in reality it can be physiotherapy intervention (up to 6 sessions) plus on single advice session.
---

	Page 5, line 139: should you refer to cost-effectiveness in your title as well and aims? Page 5 “quantify and draw inference”: this expression seems rather generic, not specific. Your aim is to compare, assess which arm is more effective than the other for improving functional status, etc. I suggest you revising this to make it very explicit to the reader what your aim is. Page 5, line 143: why not using 6-months follow-up for this, as you have for other outcomes? Page 5, line 149, “and facilitators”: and identify facilitators and barriers? Should you include the term “identify”? Page 5, SWAT: your introduction makes no reference to potential challenges with response rates and follow-up. Recruitment and retention of participants within study is always a challenge, but I think you should explicit present a rationale for this SWAT in your intro. Is there any evidence that this group of patients are challenging to recruit and follow-up? Page 5, line 226: apologies if I missed something. It is not clear the rationale and reasoning for assessing the cost of phone call and postcard for participants. This seems out of context with the SWAT research question. Line 269: shouldn't participants be randomized after being assessed for eligibility and agreeing to take part in the study? What is the rationale for allocating them after they have received part of the intervention (i.e. the first advice session)? Is there a risk of selection bias as a consequence of when randomization occurs? Line 287: is “prejudice” the most appropriate term here? Page 9, line 295: I would suggest you presenting some evidence for supporting the content of interventions planned for the trial. Where these based on best-evidence? Clinical expertise? How were these interventions and their content developed? What was the rationale for including contents described from lines 308 to 315? Also, why did you not use/follow the TIDieR checklist? Lines 341 to 344: I suggest you to elaborate and present some reference supporting your decision for patients seeking further treatment not being considered as deviation from protocol. Lines 367 to 368: what is meant by appropriate action? Are you planning an active monitoring of treatment fidelity so that, in case of protocol deviation, clinicians receive feedback and, potentially, further training to ensure treatment fidelity? This section could be expanded to enhance clarity. Line 368: what is meant by “escalated”? What actions will be performed? Line 375: will assessor be blinded? That is not explicit.
--	--

	Line 389: Which are the “oversight committees”? Line 418: I suggest you being specific about “pre-defined progression criteria.” Line 421: This statement suggests your statistical analysis plan will be developed later – which does not seem to be the case, since you present your statistical analysis plan later in your manuscript. Line 436: reference to support that? What are the strengths and limitations of your trial? I suggest discussing this in detail.
--	---

VERSION 1 – AUTHOR RESPONSE

Reviewer(s)' Comments to Author:

Reviewer: 1

- Please consider rewording "People sustaining T ASD have ongoing pain, disability, and substantial morbidity linked to high recurrence rates and subsequent need for repeated episodes of management " to link with the next sentence that seems to indicate more that we don't have these data?

RESPONSE: The abstract introduction has been reworded and clarified to focus on the primary aim of clinical effectiveness.

Introduction

- Please change males to men and females to women.

RESPONSE: Changed accordingly.

- Please reword to investigate your hypothesis "This protocol outlines the research to provide evidence regarding the nature and extent of what physiotherapy is required for the management of patients following T ASD with an embedded ‘Study Within A Trial (SWAT)’ to compare two strategies for retention of participants in RCTs.

RESPONSE: Has been re-worded.

- Is there blinding - will groups know about the other group?

RESPONSE: There is no blinding, as described on page 12 under the heading ‘blinding’

- How will you define significant neurovascular complications?

RESPONSE: This is as per local clinical practice in this pragmatic trial design.

- This is a little confusing to me -is it possible you may confound your control group?
"Participants randomised to the advice session only, will only receive this physiotherapy intervention. However, if they are experiencing ongoing issues which are not improving, as part of the control intervention, the advice is for the participant to contact the clinic they

attended or see their GP. If a participant follows this advice and self-initiates further physiotherapy this will not be deemed a deviation from the protocol. "

RESPONSE: This is part of the pragmatic design, in the control group patients 'self-initiate' further sessions if required. Any further sessions are being collected as part of the trial data set, which has been clarified in this section.

- Do people in the physio group choose how many sessions they receive?

RESPONSE: As per standard practice the number of additional physiotherapy sessions is a joint decision between the clinician and patient. This has been clarified in the text.

- In addition to whether participants see other clinicians, will you ask about medication use?

RESPONSE: medication use is captured in the health economic resource use questionnaires.

Reviewer: 2

- There are additional concerns that a single session will be compared with only 2 sessions of therapy. There are further concerns in the limitation of 6 sessions. This study therefore has no global impact as in many other countries people with a T ASD receive between 16-50 sessions of therapy.

RESPONSE: There is no limit on the number of sessions, this was an error in the abstract only that has been amended.

- I appreciate that one of the outcomes is to assess the cost effectiveness of this study. However, when compared with surgical intervention, the cost of therapy would be cheaper. My concern is that the finding of this study will present the surgical community with rationale as to why surgery should be offered to people with a FT ASD, when adequate therapy has not been fully explored.

RESPONSE: Health economic analysis will be conducted from an NHS and personal social services perspective, with methods adhering to the recommendations of the NICE Reference Case. Therefore the analysis goes beyond the cost of the interventions.

- Did the authors consider excluding greater tuberosity fractures? These are known to decrease risk of recurrence and may bias the findings. Similarly, will 'neurovascular complications' exclude those with axillary nerve palsy? These too will decrease rates of recurrence.

RESPONSE: If a greater tuberosity fracture is present, but first line management is non-operative then yes patients will be included.

- Line 69- It is unclear, why the authors chose to stratify on hand dominance, compared to other risk factors that are more widely published? Age is a known risk factor and all the evidence is in agreement. There is little agreement of hand dominance.

RESPONSE: Age is included as a stratification. Hand dominance was also chosen because the primary outcome measure is a functional score.

- Line 251 – it is unclear who decides whether the patient will be managed non-operatively? Is there the possibility of bias because this is not clearly defined?

RESPONSE: We have clarified that the treating clinician and patient will jointly decide to have surgery or not. In the UK the majority of first time T ASD patients are managed non-operatively.

- Line 304 – it is unclear how long participants will receive immobilisation? I assume this is in internal rotation but this is not made clear.

RESPONSE: We have clarified that the immobilisation will be in internal rotation, as per UK national guidelines, for a duration of up to two weeks from date of injury

- Line 346 – Again concern that 1 session will be compared to 2 sessions of therapy – Was it not possible to only have the option of 1 vs. 6?

RESPONSE: This is a pragmatic RCT, and after consultation with UK physiotherapy departments regarding current practice and in keeping with UK guidelines the comparator is the offer of at least one additional physiotherapy session. Each additional session will last for up to 30 minutes, over a maximum duration of four months from date of randomisation. We do not stipulate an upper limit on number of sessions.

- Line 349 – I am unable to access the ‘core set ‘ of generalised shoulder exercises offered to the control arm, and so cannot comment in any great depth here.

RESPONSE: The ‘core set’ is described on page 10.

Reviewer: 3

- Page 4, lines 133 to 135: should you simplify and state you are assessing the effectiveness of physiotherapy intervention plus advice compared to advice alone?

RESPONSE: We are unable to change the primary outcome for the trial in progress.

- Page 4, line 134, “physiotherapy with a single session of advice only”: I wonder whether this should be re-worded. The term “physiotherapy” refers to a profession. You are referring here to physiotherapy intervention/exercise therapy + advice. The way it is written may be misleading, as it seems it is a single physiotherapy session with advice only, when in reality it can be physiotherapy intervention (up to 6 sessions) plus on single advice session.

RESPONSE: Physiotherapy refers to an intervention not a profession in this context.

- Page 5, line 139: should you refer to cost-effectiveness in your title as well and aims?

RESPONSE: cost-effectiveness is a secondary objective detailed on page 5.

- Page 5 “quantify and draw inference”: this expression seems rather generic, not specific. Your aim is to compare, assess which arm is more effective than the other for improving functional status, etc. I suggest you revising this to make it very explicit to the reader what your aim is.

RESPONSE: The detail pertaining to how we will quantify and draw inferences is detailed in the statistical and health economic analysis sections.

- Page 5, line 143: why not using 6-months follow-up for this, as you have for other outcomes?

RESPONSE: The OSIS is the primary outcome at six month. The other outcome points are secondary objectives which is why OSIS is not mentioned here. It is being collected at six months, but as part of the primary objective.

- Page 5, line 149, “and facilitators”: and identify facilitators and barriers? Should you include the term “identify”?

RESPONSE: Added ‘identify’ accordingly.

- Page 5, SWAT: your introduction makes no reference to potential challenges with response rates and follow-up. Recruitment and retention of participants within study is always a challenge, but I think you should explicit present a rationale for this SWAT in your intro. Is there any evidence that this group of patients are challenging to recruit and follow-up?

RESPONSE: We do not anticipate this group of participants to be challenging to follow up. The SWAT is incorporated to optimise processes for future trials.

- Page 5, line 226: apologies if I missed something. It is not clear the rationale and reasoning for assessing the cost of phone call and postcard for participants. This seems out of context with the SWAT research question.

RESPONSE: The secondary SWAT objectives are ‘Cost of intervention per participant’.

- Line 269: shouldn’t participants be randomized after being assessed for eligibility and agreeing to take part in the study? What is the rationale for allocating them after they have received part of the intervention (i.e. the first advice session)? Is there a risk of selection bias as a consequence of when randomization occurs?

RESPONSE: Randomisation after the first session is designed to reduce bias.

- Line 287: is “prejudice” the most appropriate term here?

RESPONSE: Yes this is the most appropriate.

- Page 9, line 295: I would suggest you presenting some evidence for supporting the content of interventions planned for the trial. Where these based on best-evidence? Clinical expertise? How were these interventions and their content developed? What was the rationale for including contents described from lines 308 to 315?

RESPONSE: A detailed ‘intervention development’ paper is planned as a publication output. This is the protocol paper for the main trial, so does not contain this detail.

- Also, why did you not use/follow the TIDieR checklist?

RESPONSE: We are using TIDieR, as detailed on page 9.

- Lines 341 to 344: I suggest you to elaborate and present some reference supporting your decision for patients seeking further treatment not being considered as deviation from protocol.

RESPONSE: As detailed on page 11, the control treatment includes patient self-initiated further physiotherapy. Therefore if a patient initiates further sessions it is not a deviation from the protocol, it is pre-defined.

- Lines 367 to 368: what is meant by appropriate action? Are you planning an active monitoring of treatment fidelity so that, in case of protocol deviation, clinicians receive feedback and, potentially, further training to ensure treatment fidelity? This section could be expanded to enhance clarity.

RESPONSE: Yes, that's exactly what we mean and we have clarified this in the text.

- Line 368: what is meant by "escalated"? What actions will be performed?

RESPONSE: We have clarified that they will be escalated to oversight committees.

- Line 375: will assessor be blinded? That is not explicit.

RESPONSE: There is no blinding in this RCT.

- Line 389: Which are the "oversight committees"?

RESPONSE: These are detailed on page 13, documenting '...independent Trail Steering Committee (TSC) and Data Monitoring and Safety Committee (DMSC). The TSC and DMSC will hold meetings annually and be responsible for approval of the protocol, monitoring and supervising progress.'

- Line 418: I suggest you being specific about "pre-defined progression criteria."

RESPONSE: The pre-defined progression criteria have been added.

- Line 421: This statement suggests your statistical analysis plan will be developed later – which does not seem to be the case, since you present your statistical analysis plan later in your manuscript.

RESPONSE: The protocol paper presents an outline of the statistical analysis. The statistical team produce a more in depth document (SAP: Statistical Analysis Plan), which is reviewed and approved by the independent oversight committees prior to final data analysis.

- Line 436: reference to support that?

RESPONSE: In this paragraph we have presented rational for why we believe a therapist effect is unlikely in this RCT design. There is no strong evidence to support this rational (so no references), which is why we have included an interim analysis.

- What are the strengths and limitations of your trial? I suggest discussing this in detail.

RESPONSE: Main strengths and limitations have been reiterated in an additional section at the end of the paper.

VERSION 2 – REVIEW

REVIEWER	Ebonie Rio
----------	------------

	La Trobe University Australia
REVIEW RETURNED	25-Sep-2020

GENERAL COMMENTS	Thank you for considering my comments and making changes as appropriate.
--

REVIEWER	Margie Olds Flawless Motion Ltd New Zealand
REVIEW RETURNED	04-Oct-2020

GENERAL COMMENTS	I have no further comments
----------------------------

REVIEWER	Daniel Ribeiro University of Otago, New Zealand
REVIEW RETURNED	11-Oct-2020

GENERAL COMMENTS	Thank you for your responses.
-------------------------------